# Experimental Study on the Effects of Tapioca Starch on Cement Mortar Quality Improvement

**DOI:** 10.3390/ma17163889

**Published:** 2024-08-06

**Authors:** Chang-Hwan Jang, Yong-Jic Kim, Sung-Rok Oh

**Affiliations:** 1Division of Smart Construction and Environmental Engineering, Daejin University, Pocheon 11159, Republic of Korea; cjang@daejin.ac.kr; 2Department of Civil Engineering, Semyung University, Jecheon 27136, Republic of Korea

**Keywords:** cement, mortar, quality, tapioca starch

## Abstract

In this study, the effect of tapioca starch (TP) on mortar was evaluated by incorporating TP into the mortar mixture. The evaluation involved analyzing the mortar’s quality characteristics, performance, and fundamental quality improvements. The addition of TP resulted in a decrease in flow, which was attributed to increased viscosity. Specifically, a 10% reduction in flow was observed with a 0.025% increase in TP content. After 28 days, the impact of TP on the compressive strength of the mortar remained consistent, regardless of the TP amount. However, within the first 3 days, higher TP content accelerated strength development, with early compressive strength increasing by up to 20% at a 0.050% TP level. Additionally, bond strength improved by approximately 60% at a 0.050% TP concentration, and final shrinkage was reduced by 5%.

## 1. Introduction

According to recent studies, it is desirable to extend the service life of structures to reduce the environmental burden by introducing the lifecycle analysis system of concrete structures to structure evaluation [1,2,3,4]. Due to the need to maintain and preserve existing structures, the concepts of repair and reinforcement have gained new recognition. Repair materials that can be effectively used in old concrete structures are extremely limited, and cement mortar (inorganic) with properties similar to the original concrete is commonly used. To properly apply such repair materials to old structures, it is necessary to first select materials with similar adhesion, contraction, and thermal expansion properties to integrate their behavior with existing structures. Additionally, repair materials with high durability must be used to extend the service life of structures, which is the fundamental purpose of repair. Therefore, to improve the quality of repairs, it is important to use appropriate repair materials according to the type and characteristics of the structure.

Due to the development of eco-friendly and diverse repair mortar materials, studies aimed at improving the quality of repair mortars and commercialized products have been increasing [5,6,7]. In relation to this, technological advancements have been made using industrial byproducts, such as ground-granulated blast furnace slag and fly ash. Recently, there has been a push towards the diversification of chemical admixtures and additives. These chemical admixtures and additives are used to control the quality of mortar and concrete, enhancing their quality, strength, fire resistance, crack control, low density, and workability. The effect of these mixtures on the properties of concrete varies depending on several factors, such as functional composition, chemical composition, and molecular weight. For example, in the study “A Novel Approach for 3D Printing Fiber-Reinforced Mortars”, innovative methods were explored to enhance the performance of mortars using 3D printing technology and fiber reinforcement. Additionally, the study “Microstructure and Mechanical Properties of Cost-Efficient 3D Printed Concrete Reinforced with Polypropylene Fibers” examined how the incorporation of polypropylene fibers affects the structural integrity and economic feasibility of 3D printed concrete [8,9].

In Southeast Asia, tapioca starch (TP) has been used as an additive to improve the quality and performance of repair mortar [10,11,12]. Joseph and Xavier (2016) studied the effects of starch admixtures on the fresh and hardened properties of concrete, focusing on general starch effects rather than specifically on tapioca starch [10]. Okafor (2008) explored the potential of cassava flour as a set-retarding admixture in concrete, which differs from this study’s focus on the enhancement of early strength and bond strength using tapioca starch [11]. Wanishlamlert et al. (2017) investigated the impact of tapioca starch on self-compacting concrete, whereas this study emphasizes its effects on mortar quality characteristics, including viscosity, compressive strength, and shrinkage reduction [12].

TP is obtained by processing the starch collected from the roots of cassava, a tropical plant growing in Central America, Southeast Asia, and Africa, into powder form [13]. Previously, TP was mainly used in the food sector. As TP was known to improve the performance of repair mortar when used as an additive, studies to utilize TP as a repair mortar additive and in commercialized products have been introduced in Southeast Asia. Cases in which TP was investigated as a repair mortar additive are limited to Southeast Asia and nearby countries, and related research and technological trends are difficult to find [10,11,12]. Accordingly, this study aims to develop a domestic repair mortar with improved flow, mechanical properties, and durability by incorporating tapioca starch (TP) as an additive. To assess TP’s potential as a substitute for traditional starch-based chemical additives, prior cases where such additives were used were investigated. Following this analysis, TP was chosen, and its impact on the mortar’s flow, mechanical characteristics, and durability was examined to evaluate its effectiveness as an alternative to conventional chemical additives or thickeners.

In Korea, it is difficult to find cases where TP has been used in repair mortar. Cases of using TP in the construction sector are often found in patents, but most of these only introduce technologies for using it as an admixture or additive. Overseas, studies have been conducted mainly in regions where cassava is commonly grown. Methods to apply TP in the construction sector have been reported in South Africa and Vietnam since 2010 [14,15,16,17,18,19,20,21,22,23]. Overall, TP improves initial strength and interfacial adhesion while decreasing creep deformation by increasing the coefficient of elasticity [20].

Therefore, in this study, mortar mixed with TP was prepared to evaluate the effect of TP on mortar performance. The quality characteristics of the mortar were assessed to determine the impact of TP on performance and basic quality improvement. The results of this study are expected to serve as basic data for the application of TP in repair mortar and construction materials, such as concrete.

## 2. Experiment Outline

### 2.1. Experiment Plan Evaluation Method Selection

The use of tapioca starch (TP) as an additive in mortar is an innovative approach that has not been extensively explored in prior studies. The research demonstrates how TP, a biodegradable and eco-friendly material, can enhance the performance characteristics of mortar. The experimental plan included a comprehensive evaluation of multiple quality characteristics of mortar, including flow, compressive strength, bond strength, shrinkage, and durability properties. This holistic approach provides a more complete understanding of the impact of TP on mortar.

In this study, various evaluation items and variables were used to comprehensively assess the quality and performance of cement mortar mixed with tapioca starch (TP). The evaluation standards for each item are listed in Table 1. Three different TP contents (0.025%, 0.050%, and 0.075%) relative to the mass of ordinary Portland cement (OPC) were tested.

The mortar mixed with tapioca starch was evaluated based on rheological properties, mechanical properties, and durability properties. For rheological properties, flow and setting time were measured using a flow table test and a Vicat needle apparatus, respectively, following ASTM standards. Mechanical properties were assessed by measuring compressive strength and flexural strength at different curing ages (3, 7, 28 days) using standardized test methods (KS L ISO 679). Durability properties were evaluated by examining drying shrinkage and carbonation depth according to ASTM guidelines.

Three representative aspects were selected for each category of evaluation. Rheological properties included flow, setting time, and viscosity. Mechanical properties included compressive strength, flexural strength, and bond strength. Durability properties included drying shrinkage, carbonation resistance, and chloride ion penetration resistance. Table 1 provides an overview of the evaluation items and variables used in this study.

Additionally, the effects of TP on the mortar’s performance were analyzed through detailed comparisons of the test results across different TP content levels. This comprehensive approach allowed for a thorough understanding of how TP influences various aspects of mortar performance, providing valuable insights for the development of high-quality repair materials.

### 2.2. Experimental Materials

#### 2.2.1. Cement Mortar

The cement used for preparing the mortar was ordinary Portland cement (OPC) with a density of 3.14 g/cm^3^. Silica sand (S) with a density of 2.54 g/cm^3^ was used as fine aggregate. Table 2 shows the chemical components and physical properties of OPC and Table 3 lists the physical properties of S. Table 4 shows the mix proportions of mortar according to the TP content.

#### 2.2.2. TP

The particle diameter of tapioca starch (TP) used in the experiment ranged from 4 to 35 μm. Figure 1 illustrates the cassava roots, which are the raw material for TP. The cassava roots undergo a series of processing steps to extract the TP, as shown in Figure 2. This includes washing, peeling, grinding, and drying to obtain the final powdered form of TP. Table 5 provides the physical properties of TP, including its particle size distribution, moisture content, and density. This detailed characterization of TP is essential for understanding its interaction with cement mortar and its subsequent impact on the mortar’s properties.

### 2.3. Experimental Method

#### 2.3.1. Rheological Properties

Flow

The flow of mortar was measured in accordance with “ASTM C 1437-20 Standard Test Method for Flow of Hydraulic Cement Mortar”. This standard outlines the procedure for determining the flow of hydraulic cement mortar using a flow table. The mortar is placed on a flow table, which is then dropped a specified number of times to allow the mortar to spread. The resulting diameter of the spread mortar is measured and recorded as the flow value. This method provides an indication of the workability and consistency of the mortar.

2.Setting

The setting of mortar was performed in accordance with “ASTM C 191-21 Standard Test Methods for Time of Setting of Hydraulic Cement by Vicat Needle”. This standard specifies the method for determining the initial and final setting times of hydraulic cement using a Vicat needle. The mortar is placed in a mold and the Vicat needle is used to penetrate the surface at regular intervals. The initial setting time is recorded when the needle ceases to penetrate beyond a specified depth, and the final setting time is recorded when the needle no longer leaves an impression on the mortar surface. These measurements are crucial for understanding the workability and setting behavior of the mortar.

3.Rheology

The rheology of the mortar was evaluated by referring to the results of previous studies due to the lack of related regulations [24,25]. The rheology measurements were conducted using a Brookfield DV-III Ultra mortar viscometer with a modified chamber size. Generally, the fluidity of mortar is determined through a flow test, but flow test results often fail to adequately represent the rheological properties of semi-plastic mortar. Rheological properties are related to the workability and viscosity of the mortar.

Typically, the viscosity of fluids, such as water and oil, is characterized using a Newtonian model, as shown in Figure 2. However, because mortar is not in a purely liquid phase, it is better represented by the Bingham model, which indicates that it does not flow until a certain external force is applied. The Bingham model can be obtained by correlating the shear rate and shear stress measured using a viscometer. The rheological properties of mortars were evaluated based on the plastic viscosity and yield stress, which are represented by the slope and y-intercept of the Bingham model, respectively [24,25].

#### 2.3.2. Mechanical Properties

Compressive Strength

The compressive strength of mortar was measured in accordance with “KS L ISO 679 Methods of Testing Cements-Determination of Strength”. This standard specifies the method for determining the compressive strength of hydraulic cement and other mortars, which involves preparing and curing specimens under controlled conditions. The specimens are then subjected to a compressive force until failure, and the maximum load is recorded. The compressive strength is calculated by dividing the maximum load by the cross-sectional area of the specimen.

2.Flexural strength

The flexural strength of mortar was measured in accordance with “KS L ISO 679 Methods of Testing Cements-Determination of Strength”. This standard outlines the procedure for determining the flexural strength of hydraulic cement and other mortars. Specimens are prepared and cured under controlled conditions and are then subjected to a bending force until failure. The maximum load at failure is recorded, and the flexural strength is calculated using the formula specified in the standard, which accounts for the dimensions of the specimen and the applied load.

3.Bond strength

The bond strength of mortar was measured in accordance with “ASTM C1585/C1583M-20 Standard Test Method for Tensile Strength of Concrete Surfaces and the Bond Strength or Tensile Strength of Concrete Repair and Overlay Materials by Direct Tension (Pull-off Method)”. This standard describes the procedure for evaluating the tensile bond strength of concrete surfaces and the bond strength or tensile strength of concrete repair and overlay materials. Specimens are prepared by attaching a loading fixture to the surface of the mortar and applying a direct tensile force until failure. The maximum load at failure is recorded, and the bond strength is calculated by dividing this load by the cross-sectional area of the failure surface.

#### 2.3.3. Durability Properties

Drying shrinkage

For the drying shrinkage of mortar, the length change was measured in accordance with “ASTM C157/C157M-17 Standard Test Method for Length Change of Hardened Hydraulic-Cement Mortar and Concrete”. This standard specifies the procedure for determining the length change of mortar specimens subjected to drying conditions. Specimens are cast and cured under standardized conditions and then placed in a controlled environment where the relative humidity and temperature are maintained. The length change of the specimens is measured at regular intervals to determine the drying shrinkage. This measurement is critical for evaluating the potential for cracking and durability of the mortar under drying conditions.

2.Carbonation

For the carbonation resistance of mortar, the carbonation depth was measured in accordance with “ASTM C1905/C1905M-23 Standard Specification for Cements that Require Carbonation Curing”. This standard outlines the procedure for determining the carbonation depth of mortar specimens subjected to controlled carbonation conditions. Specimens are prepared and cured under standardized conditions and are then exposed to a carbon dioxide-rich environment. The carbonation depth is measured at specified intervals using a phenolphthalein indicator solution, which reacts with the carbonated areas of the mortar. This measurement is essential for assessing the durability and long-term performance of the mortar in environments where carbonation is a concern.

3.Chloride ion penetration resistance

The chloride ion penetration resistance of mortar was evaluated in accordance with “ASTM C 1202-22e1 Standard Test Method for Electrical Indication of Concrete’s Ability to Resist Chloride Ion Penetration”. In this method, the diffusion cell anode (+) was filled with a 0.3 N NaOH solution and the cathode (−) was filled with a 3% NaCl solution. A voltage of 30 V was applied to both ends of the specimen for 8 h. After the test, the specimen was cut, and a 0.1 N AgNO₃ solution was sprayed onto the cut surface to measure the penetration depth using the chloride colorimetric method. The diffusion coefficient was estimated using Equation (1). This method allows for the rapid measurement of the chloride diffusion coefficient in concrete using an electric field [26].
(1)$$D=\frac{RTL}{xFU}\cdot \frac{x_{d}-\alpha \sqrt{x_{d}}}{t}$$
where
$$\alpha =2\sqrt{\frac{RTL}{zPU}}{erf}^{-1}\left(1-\frac{2C_{d}}{C_{0}}\right)$$



$D:\;Chloride\;ion\;diffusion\;coefficient\left(m^{2}/s\right)$



$R:Gas\;constant\left(8.314\;J/K\cdot mol\right)$



$T:Temperature\;of\;the\;solution\left(K\right)$



$L:Specimen\;thickness\left(m\right)$



$z:Ion\;valence\left(z=1\;for\;chloride\;ions\right)$



$F:Faraday\;constant\left(96,481.04\;J/V\cdot mol\right)$



$U:Potential\;difference\;between\;the\;anode\;and\;cathode\left(V\right)$



$x_{d}:Chloride\;ion\;penetration\;depth\left(m\right)$



$t:Experiment\;duration\left(h\right)$



erf:Error function



$C_{0}:Chloride\;concentration\;in\;the\;cathode\;solution\left(mol/L\right)$



$C_{d}:Chloride\;concentration\;at\;the\;color\;change\;position\left(mol/L\right)$



## 3. Experimental Results and Discussion

### 3.1. Effect of TP on the Rheological Properties of Cement Mortar

#### 3.1.1. Rheology

Figure 3 illustrates the rheology measurement results of mortar mixed with tapioca starch (TP), and Table 6 lists the plastic viscosity and yield stress according to the TP content. The results indicate that the plastic viscosity tends to increase while the yield stress tends to decrease as the TP content increases. This suggests that the workability of the mortar decreases with higher TP content due to the increased viscosity. The rheological properties are critical for understanding the workability and application of mortar in construction. The increase in plastic viscosity with higher TP content implies that the mortar becomes more resistant to flow, making it harder to work with. Conversely, the decrease in yield stress suggests that less force is required to initiate flow in the mortar, which could be beneficial in certain applications where initial flow is crucial. These findings highlight the trade-off between viscosity and yield stress in the design of mortar mixtures, emphasizing the need for a balanced approach to achieve optimal workability.

#### 3.1.2. Flow

Table 6 shows the flow measurement results for mortar mixed with tapioca starch (TP). It can be seen that the mortar flow tended to decrease proportionally as the TP content increased at the same water-to-cement (W/C) ratio. Compared to the flow of plain mortar, the flow values of TP1, TP2, and TP3 decreased by 10%, 20%, and 30%, respectively.

According to previous studies, when the viscosity of dough is measured using a Farinograph, TP exhibits 1000 Brabender units (BU), whereas corn starch shows a relatively lower value of 600 BU. This indicates that TP, with its higher viscosity, is more effective in absorbing moisture, leading to enhanced gelation compared to corn starch, despite their similar starch properties [15,16,17]. In the case of mortar, TP reduces the flow by absorbing water initially, but it has been reported that the gelation effect of TP can improve quality by filling the micropores of the mortar. However, the gelation phenomenon of TP is considered to be highly sensitive because it significantly decreases the flow by restricting mortar, and it is necessary to add TP at a level that meets the required flow. According to the results of this study, it is possible to secure workability with a TP content of up to 0.075% at the same W/C ratio.

#### 3.1.3. Setting

Table 7 shows the initial and final setting time measurement results for mortar mixed with tapioca starch (TP). Figure 4 illustrates the results of the penetration resistance test. The initial setting time was found to be 4 h for plain mortar, 4 h and 10 min for TP1, and 4 h and 15 min for TP2 and TP3. The final setting time was 4 h and 20 min for plain mortar, 4 h and 32 min for TP1, 4 h and 40 min for TP2, and 4 h and 43 min for TP3.

These results indicate that TP delays the initial setting time of repair mortar by 10 to 15 min and the final setting time by 12 to 23 min. According to previous studies, starch may serve as a retardant because it has a hardening delay effect [15]. TP is expected to improve the work time during construction by enhancing the delay performance of repair mortar.

### 3.2. Effect of TP on the Mechanical Properties of Cement Mortar

#### 3.2.1. Compressive Strength

Table 7 summarizes the results of evaluating the mechanical properties of mortar mixed with tapioca starch (TP), and Figure 5 shows the compressive strength improvement rates of TP1, TP2, and TP3 compared to that of plain mortar.

According to the compressive strength measurement results shown in Table 7, the compressive strength of mortar was identical at 28 days regardless of the TP content. The initial compressive strength at 3 days increased proportionally with the TP content. Though it tended to decrease at a TP content of 0.075%, it remained equal to the compressive strength of plain mortar. Compressive strength at 7 days also showed the same tendency as at 3 days. Figure 5 illustrates that the compressive strength improvement rate at the early ages of 3 and 7 days increased as the TP content increased. However, when the TP content exceeded a certain level, its effect did not increase proportionally.

These results indicate that there is an optimal TP content, suggesting that the optimal TP content can enhance the strength improvement rate while maintaining the required flow. Therefore, when an appropriate amount of TP is mixed with mortar, it is expected to improve the compressive strength at the early ages of 3 and 7 days. However, the compressive strength at 28 days remained identical regardless of the TP content. This suggests that TP does not affect long-term strength after 28 days, although it significantly impacts the development of initial strength. It appears that TP improves the initial strength by filling the internal pores due to the initial gelation effect.

#### 3.2.2. Flexural Strength

The flexural strength measurement results in Table 8 show that the flexural strength tended to increase for TP1 and TP2, similar to the trend observed in compressive strength, but the flexural strength of TP3 was equal to that of plain mortar. At 28 days, the flexural strength of plain mortar was 7.8 MPa, and those of TP1, TP2, and TP3 were approximately 7.9 MPa. These values are considered to be at the same level, as there is no significant difference from the flexural strength of the existing repair mortar.

It is concluded that TP has no significant impact on the flexural strength of repair mortar, and the use of TP does not provide a flexural strength improvement effect due to the very narrow strength range observed.

#### 3.2.3. Bond Strength

As observed from Table 8, the bond strength tended to increase proportionally as the TP content increased, and it decreased to the same level as plain mortar after the TP content exceeded a certain level. This trend is similar to the tendency observed in compressive strength at early ages. It is concluded that the bond strength increased with a TP content of up to 0.050% (TP2), but the effect of TP decreased at 0.075% (TP3). These results indicate that TP, a starch-based material, enhances the bond level at the interface by increasing the viscosity when mixed at an optimal level, thereby improving adsorption performance.

For the bond strength of typical repair mortar, a value of 1 MPa or higher is required without the application of primer to the old surface (KS F 4042). The plain mortar used in this study developed a bond strength of approximately 1.8 MPa. When TP was mixed, the bond strength improved by up to 60%. As approximately three times higher bond strength is secured compared to the typical repair mortar standard, the addition of an appropriate level of TP is expected to exhibit a significant bond strength improvement effect. However, it is crucial to apply the optimal TP content because the effect of TP diminishes beyond a certain level, similar to the trend observed in compressive strength. Due to the low content and high sensitivity, it is highly important to select the appropriate range of TP content for quality control.

### 3.3. Effect of TP on the Durability Properties of Cement Mortar

#### 3.3.1. Drying Shrinkage (Length Change)

The length change measurement results in Table 9 show that the final shrinkage decreased by approximately 5% due to the addition of tapioca starch (TP). At early ages, the expansion effect tended to double for up to 3 days but converged to a certain value after 7 days. This phenomenon appears to be caused by the optimal TP content, which is consistent with previous evaluation results and analysis.

It is concluded that the high swelling and absorption power of TP leads to a shrinkage compensation effect caused by the expansion of TP. This indicates that it is possible to compensate for shrinkage with the initial expansion effect by rapidly absorbing the surplus water in the mortar in the early age.

#### 3.3.2. Carbonation

Table 9 shows the carbonation depth measurement results for mortar mixed with tapioca starch (TP). Figure 6 illustrates the tendency of carbonation resistance. After 4 weeks, the carbonation depth according to the TP content was found to be 0.83, 0.66, 0.55, and 0.39 mm for plain, TP1, TP2, and TP3, respectively. These values changed to 0.90, 0.72, 0.63, and 0.44 mm after 16 weeks. These results indicate that TP with high fineness increased carbonation resistance by densely filling the internal pores of the mortar, thereby reducing the penetration of carbon dioxide.

#### 3.3.3. Chloride Ion Penetration Resistance

Figure 7 shows the chloride ion penetration resistance evaluation results for mortar mixed with tapioca starch (TP), whereas Table 10 lists the chloride ion diffusion coefficient results. The results of Figure 7 indicate that the chloride ion penetration depth of mortar mixed with TP slightly decreased compared to that of plain mortar. However, the difference is minimal, suggesting that TP has no significant impact on chloride ion penetration resistance.

In Table 10, the chloride ion diffusion coefficient decreased in the order of TP1 > TP2 > TP3 > Plain. However, there was no significant difference in the TP content, considering the error range of the total charge and diffusion coefficient. As the addition of TP decreased the diffusion coefficient compared to that of plain mortar, it is concluded that chloride ion penetration resistance was improved. This improvement is attributed to TP enhancing the density of the internal matrix through gelation inside the repair mortar, thus reducing the chloride ion diffusion coefficient of plain mortar.

## 4. Conclusions

In this study, mortar mixed with tapioca starch (TP) was prepared to evaluate the effect of TP on mortar, and its quality characteristics were assessed. The following conclusions were drawn:When TP was mixed with mortar, the flow tended to decrease due to the increase in viscosity of the fresh mortar. The flow decreased by up to 30% as the TP content increased by 0.025%.The impact of TP on the compressive strength of mortar was found to be identical regardless of the TP content at 28 days. However, strength development was accelerated within 3 days depending on the TP content.The bond strength of mortar improved by approximately 60% when the TP content was 0.050%, achieving approximately three times higher performance compared to the required performance of repair mortar.The final shrinkage of mortar decreased by 5% due to the addition of TP, and the expansion effect doubled at early ages for up to 3 days.The durability properties of mortar, including carbonation depth and chloride ion penetration resistance, improved as the TP content increased. The chloride ion diffusion coefficient decreased with the addition of TP, indicating enhanced durability.

These results indicate that TP accelerates strength development at early ages and increases bond strength due to its properties. Additionally, shrinkage compensation through the expansion effect is considered possible. Based on the results of this study, the optimal TP content is judged to be 0.050%. TP is expected to improve the quality performance of mortar. Due to its high sensitivity even at a very low content, further research is necessary to derive utilization measures and select an appropriate range of use for quality control.

The findings regarding the optimal TP content (0.050%) provide valuable insights for practical applications in construction, suggesting that even a small addition of TP can significantly improve the quality of mortar without compromising its performance. This study contributes to the field by demonstrating that TP can effectively reduce carbonation depth and improve chloride ion penetration resistance, which are critical factors for the durability and longevity of concrete structures. In summary, the results suggest that TP can be a viable additive for improving the quality and durability of mortar, paving the way for further exploration and practical applications in the construction industry.

## Figures and Tables

**Figure 1 materials-17-03889-f001:**
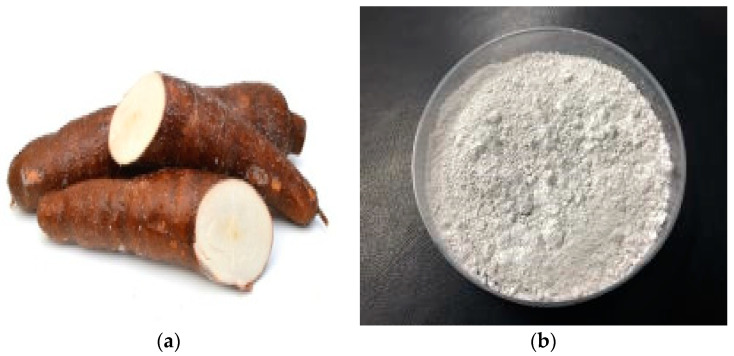
TP: (**a**) samples of cassava roots (raw material of TP), (**b**) dried and crushed TP powder sample.

**Figure 2 materials-17-03889-f002:**
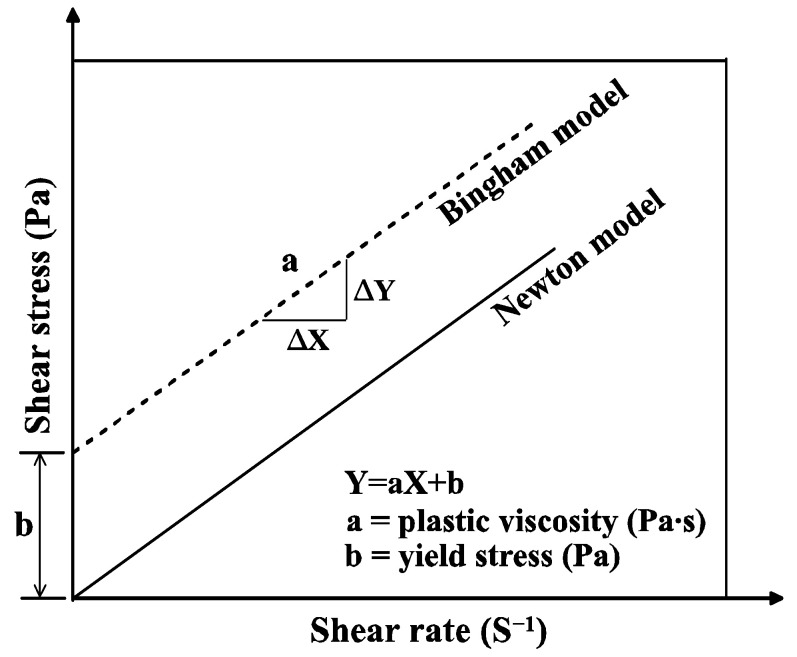
Rheology analysis model.

**Figure 3 materials-17-03889-f003:**
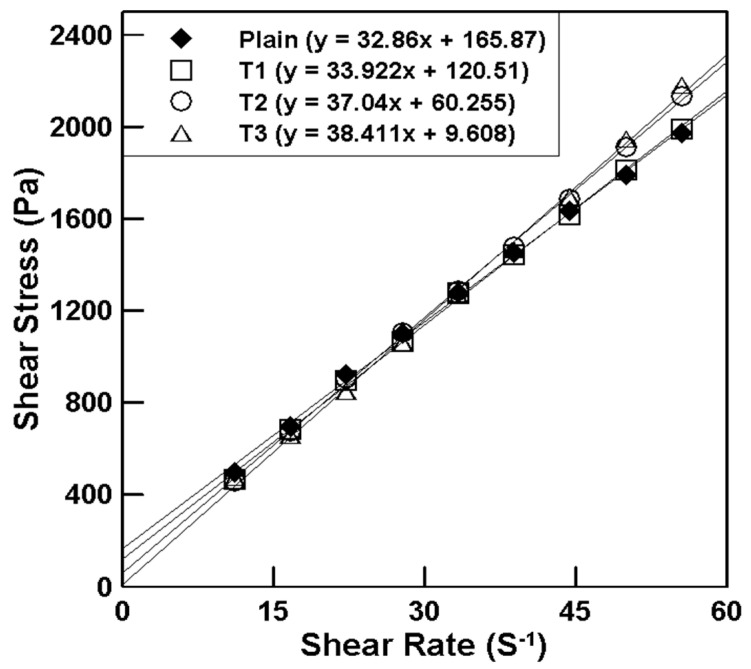
Relationship between the shear rate and shear stress for various TP content.

**Figure 4 materials-17-03889-f004:**
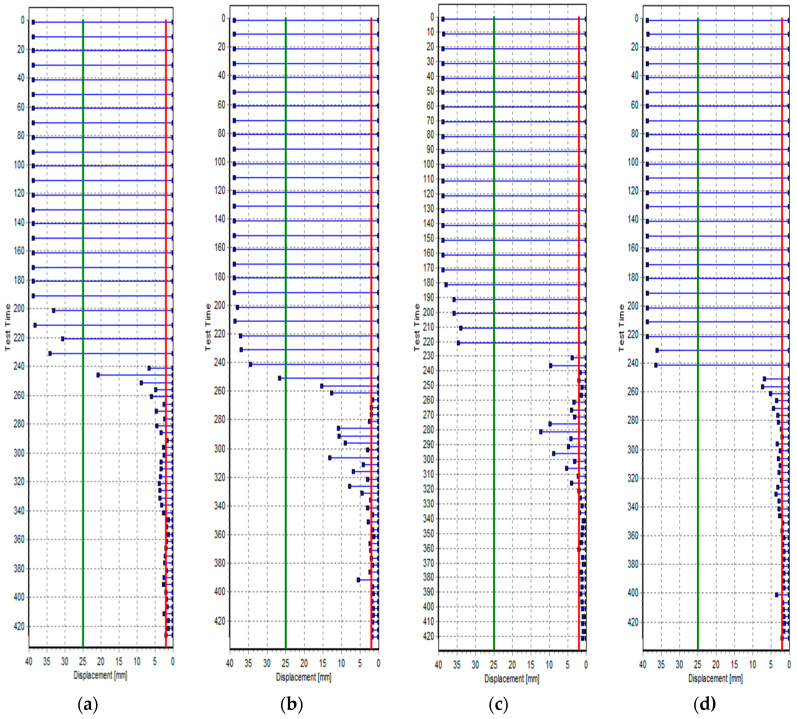
Penetration resistance test results: (**a**) Plain, (**b**) TP1, (**c**) TP2, and (**d**) TP3. The green line is the initial setting line, and the red line is the final setting line.

**Figure 5 materials-17-03889-f005:**
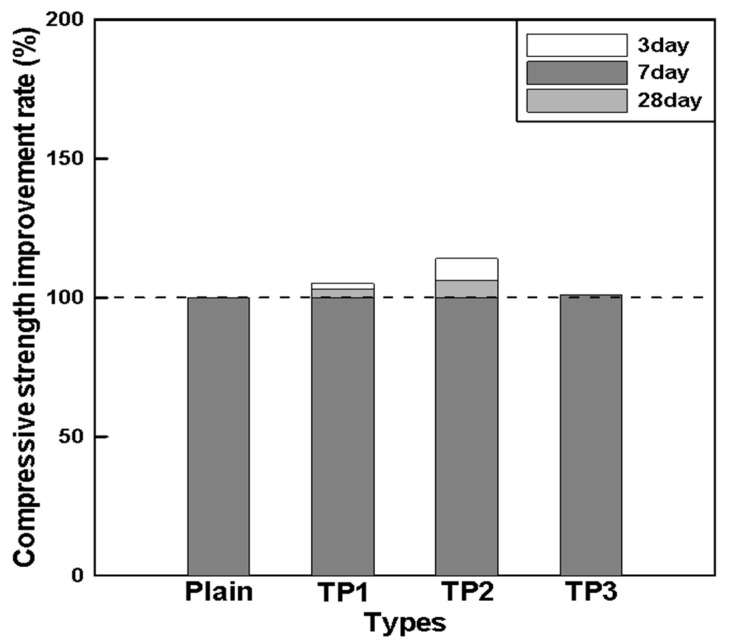
Relationship between the TP content and strength improvement rate.

**Figure 6 materials-17-03889-f006:**
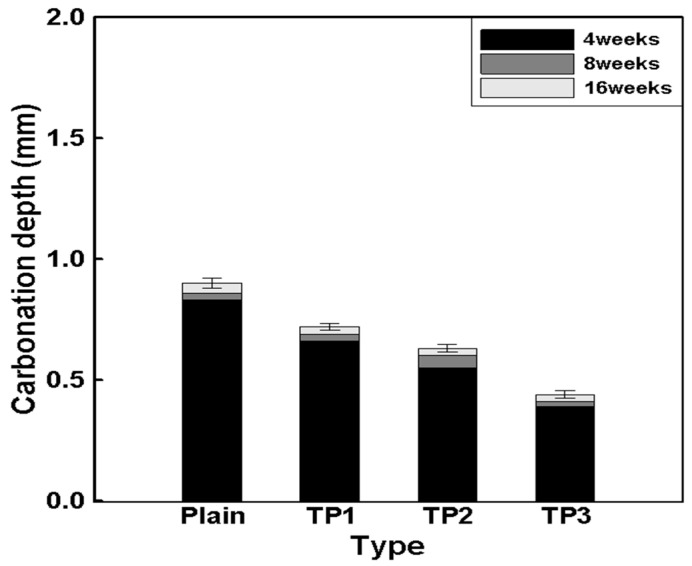
Relationship between the TP content and carbonation depth according to the TP carbonation age.

**Figure 7 materials-17-03889-f007:**
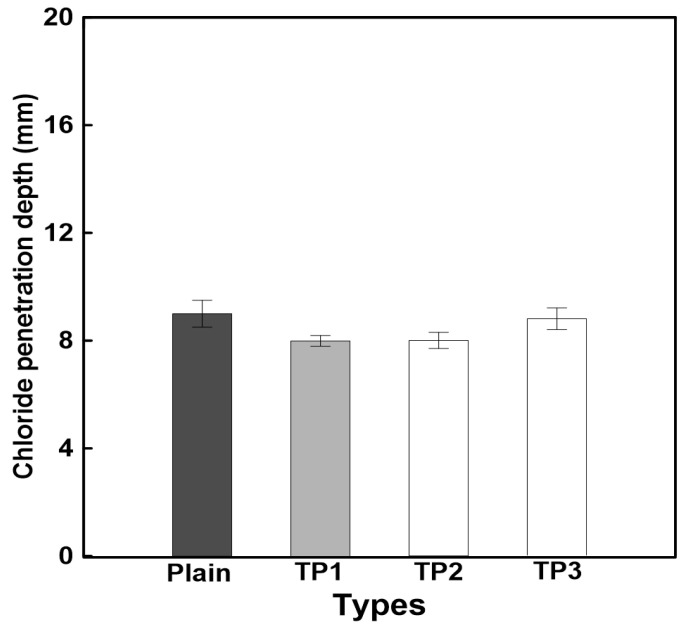
Chloride ion penetration resistance evaluation results.

**Table 1 materials-17-03889-t001:** Evaluation items and variables.

Type	Item	Evaluation Standard
Rheological properties	Flow	ASTM C 1437-20
	Setting	ASTM C 191-21
	Rheology	Internal Evaluation Criteria Applied
Mechanical properties	Compressive strength	KS L ISO 679
	Flexural strength	KS L ISO 679
	Bond strength	ASTM C1585/C1583M-20
Durability properties	Drying shrinkage	ASTM C157/C157M-17
	Carbonation	ASTM C1905/C1905M-23
	Chloride ion penetration resistance	ASTM C 1202-22e1

**Table 2 materials-17-03889-t002:** Physicochemical properties of cement.

Type	CaO (%)	SiO_2_ (%)	Al_2_O_3_ (%)	MgO (%)	Fe_2_O_3_ (%)	SO_3_ (%)	L.O.I (%)	Surface Area (cm^2^/g)	Density (g/cm^3^)
OPC	61.40	21.60	3.40	2.50	3.10	2.50	0.03	3540	3.14

**Table 3 materials-17-03889-t003:** Physical properties of fine aggregate.

Type	Gmax (mm)	Density (g/cm^3^)	Absorption (%)	F.M.	Unit Mass (kg/m^3^)
S	-	2.54	1.17	3.02	1739

**Table 4 materials-17-03889-t004:** Mix proportions of mortar with respect to the TP content.

No.	Water (W)	Binder (B)	Sand (S)	Tapioca Starch (TP)
Plain	0.4	1	1.5	-
TP1	0.4	1	1.5	0.025
TP2	0.4	1	1.5	0.050
TP3	0.4	1	1.5	0.075

**Table 5 materials-17-03889-t005:** Physical properties of TP.

Type	Diameter (μm)	Avg. Diameter (μm)	Amylopectin (%)	Amylose (%)	Density (g/cm^3^)
TP	4–35	2 0	83	17	1.6

**Table 6 materials-17-03889-t006:** Test results of rheological properties.

Mix Type	Plastic Viscosity (Pa_·_s)	Yield Stress (Pa)	Flow (mm)		
			0 min	30 min	60 min
Plain	32.86	165.87	210	205	195
TP1	33.92	120.51	190	185	182
TP2 TP3	37.04 38.41	60.26 9.6	180 170	174 162	170 158

**Table 7 materials-17-03889-t007:** Initial and final setting time according to the standard penetration resistance test.

Type	Initial Setting Time	Final Setting Time
Plain	04:00	04:20
TP1	04:10	04:32
TP2	04:15	04:40
TP3	04:15	04:43

**Table 8 materials-17-03889-t008:** Test result of mechanical properties.

Mix Type	Compressive Strength (MPa)			Flexural Strength (MPa)		Bond Strength (MPa)
	3 Days	7 Days	28 Days	28 Days	28 Days	28 Days
Plain	27	31	34	7.0	7.8	1.8
TP1	28	32	34	7.3	8.0	2.0
TP2	29	33	34	7.6	8.0	3.2
TP3	27	31	34	7.1	7.9	1.7

**Table 9 materials-17-03889-t009:** Carbonation depth measurement results.

Type	Length Change (10^−6^ με)	Carbonation Depth		
	28 Days	4 Weeks	8 Weeks	16 Weeks
Plain	−800	0.83 mm	0.86 mm	0.90 mm
TP1	−750	0.66 mm	0.69 mm	0.72 mm
TP2	−740	0.55 mm	0.60 mm	0.63 mm
TP3	−790	0.39 mm	0.41 mm	0.44 mm

**Table 10 materials-17-03889-t010:** Chloride ion diffusion coefficient analysis results.

Chloride Ion Penetrability Based on Charge Passed			
Total Passed Charge (Coulombs)		Chloride Ion Penetrability	
>4000		High	
2000–4000		Moderate	
1000–2000		Low	
100–1000		Very low	
<100		Negligible	
Type	Total charge	Diffusion coefficient	Decision
Plain	2050.74	6.2351 × 10^−12^	Moderate
TP1	1831.68	5.6706 × 10^−12^	Low
TP2	1973.25	6.0366 × 10^−12^	Low
TP3	1899.99	5.8477 × 10^−12^	Low

## Data Availability

The original contributions presented in the study are included in the article, further inquiries can be directed to the corresponding authors.

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
