# Peer review of "Experimental Study on the Effects of Tapioca Starch on Cement Mortar Quality Improvement"

_materials, 2024, doi:10.3390/ma17163889_

Round 1

Reviewer 1 Report

Comments and Suggestions for Authors

Line 19 "Recently it has been suggested.." Please rephrase: "According to.." and use proper citation.

Lines 32-41. You may find additional information in:

https://doi.org/10.3390/ma16134609

https://doi.org/10.3390/buildings13112813

Figure 2 seems to be blury. Please improve the quality.

Equation 1 seems to be a captation from another document. Please revise and cite accordingly.

Figure 3 - Use the same format and font as for the rest of the text.

I recommend that the manuscript undergo English editing to enhance its clarity and readability.

Comments on the Quality of English Language

The current version contains some uncommon formulations and language issues that need to be addressed to ensure clarity and readability. An improvement in language quality will enhance the overall presentation of the research.

Reviewer 2 Report

Comments and Suggestions for Authors

1.      Tell the difference between the idea of your work with that of the Refs.4-6, since they are also focused on effect of (cassava or tapioca) starch on properties of concrete.

2.      Add the footnotes in Table 3 for explanation of the abbreviations of W and B.

3.      Cite the literature for the statement “According to 186 previous studies, starch may serve as a retardant because it has a hardening delay effect.”

Reviewer 3 Report

Comments and Suggestions for Authors

This manuscript presented the effect of tapioca starch (TP) on mortar. It is a interesting study, however, some concerns should be addresses carefully. This manuscript may be suitable for publication in Materials, but it needs major revision to the science and figures.

1. A schematic illustration is needed on the reaction/preparation of the system to help the readers to understand the process.

2. The experimental method was presented without depth discussion. Better discussion is suggested

3. The research results of this article did not express good effectiveness and innovation. It is recommended to discover the innovative points of this article from the experimental plan to the result data, and elevate the article

4. This study says nothing about the morfological o chemical information of the samples. SEM or XDR analysis should be detailed.

Comments on the Quality of English Language

Minor editing of English language required

Reviewer 4 Report

Comments and Suggestions for Authors

The work examines the influence of tapioca starch on the technological characteristics of cement mortar. The study is relevant in the context of the development of new environmentally friendly materials for construction and repair.

The article is assessed positively, but there are comments:

1. It is recommended to improve the introductory part of the article. In this part, the authors indicate that “previous cases of using starch-based chemical additives were analyzed to examine the applicability of TP with similar components as a substitute for chemical additives or thickeners” but the analysis itself is not presented in the article. The authors refer to specific previous studies, but the advantages and disadvantages of these individual studies, and what specific problem was not solved in these studies, should be highlighted.

2. In Figure 7, the ordinate axis is incorrectly labeled; there should be values for the depth of chloride penetration, and not the depth of carbonation.

Overall, the study confirms the potential of tapioca starch as an effective additive to cement mortars to improve their quality characteristics and durability, which is especially important in the context of the development of sustainable construction technologies. The material presented in the article will be of interest to specialists involved in the development of repair compositions for concrete structures.

The article can be accepted after making appropriate amendments.

Round 2

Reviewer 3 Report

Comments and Suggestions for Authors

Accept in present form